# Recent Insights into the Measurement of Carbon Dioxide Concentrations for Clinical Practice in Respiratory Medicine

**DOI:** 10.3390/s21165636

**Published:** 2021-08-21

**Authors:** Akira Umeda, Masahiro Ishizaka, Akane Ikeda, Kazuya Miyagawa, Atsumi Mochida, Hiroshi Takeda, Kotaro Takeda, Isato Fukushi, Yasumasa Okada, David Gozal

**Affiliations:** 1Department of General Medicine, School of Medicine, IUHW Shioya Hospital, International University of Health and Welfare (IUHW), Yaita 329-2145, Japan; 2Department of Physical Therapy, School of Health Science, International University of Health and Welfare, Otawara 324-8501, Japan; ishizaka@iuhw.ac.jp; 3Department of Rehabilitation, IUHW Shioya Hospital, International University of Health and Welfare (IUHW), Yaita 329-2145, Japan; 1012003@g.iuhw.ac.jp; 4Department of Pharmacology, School of Pharmacy, International University of Health and Welfare, Otawara 324-8501, Japan; miyagawa@iuhw.ac.jp (K.M.); atsusaitou@iuhw.ac.jp (A.M.); hirotakeda@iuhw.ac.jp (H.T.); 5Department of Pharmacology, School of Pharmacy at Fukuoka, International University of Health and Welfare, Okawa 831-8501, Japan; 6Faculty of Rehabilitation, School of Healthcare, Fujita Health University, Toyoake 470-1192, Japan; ktakeda@fujita-hu.ac.jp; 7Faculty of Health Sciences, Uekusa Gakuen University, Chiba 264-0007, Japan; fukushi@1998.jukuin.keio.ac.jp; 8Laboratory of Electrophysiology, Clinical Research Center, Murayama Medical Center, Musashimurayama 208-0011, Japan; yasumasaokada@1979.jukuin.keio.ac.jp; 9Department of Child Health and the Child Health Research Institute, MU Women’s and Children’s Hospital, University of Missouri, Columbia, MO 65201, USA; gozald@health.missouri.edu

**Keywords:** carbon dioxide, transcutaneous partial pressure, end-tidal partial pressure, Bland–Altman analysis, blood gas analysis

## Abstract

In the field of respiratory clinical practice, the importance of measuring carbon dioxide (CO_2_) concentrations cannot be overemphasized. Within the body, assessment of the arterial partial pressure of CO_2_ (PaCO_2_) has been the gold standard for many decades. Non-invasive assessments are usually predicated on the measurement of CO_2_ concentrations in the air, usually using an infrared analyzer, and these data are clearly important regarding climate changes as well as regulations of air quality in buildings to ascertain adequate ventilation. Measurements of CO_2_ production with oxygen consumption yield important indices such as the respiratory quotient and estimates of energy expenditure, which may be used for further investigation in the various fields of metabolism, obesity, sleep disorders, and lifestyle-related issues. Measures of PaCO_2_ are nowadays performed using the Severinghaus electrode in arterial blood or in arterialized capillary blood, while the same electrode system has been modified to enable relatively accurate non-invasive monitoring of the transcutaneous partial pressure of CO_2_ (PtcCO_2_). PtcCO_2_ monitoring during sleep can be helpful for evaluating sleep apnea syndrome, particularly in children. End-tidal PCO_2_ is inferior to PtcCO_2_ as far as accuracy, but it provides breath-by-breath estimates of respiratory gas exchange, while PtcCO_2_ reflects temporal trends in alveolar ventilation. The frequency of monitoring end-tidal PCO_2_ has markedly increased in light of its multiple applications (e.g., verify endotracheal intubation, anesthesia or mechanical ventilation, exercise testing, respiratory patterning during sleep, etc.).

## 1. Introduction

Atmospheric carbon dioxide (CO_2_) concentration is increasing worldwide by the increasing consumption of carbon-based combustibles along with progressive deforestation [1,2]. Increases in atmospheric CO_2_ concentrations are thought to cause elevation of atmospheric temperature as a result of the greenhouse effect. High concentrations of atmospheric CO_2_ can facilitate the onset of human health problems, such as increased fatigue, headache, and tinnitus. Inhalation of 0.1% CO_2_ for a short time has been reported to cause marked changes in respiratory, circulatory, and cerebral electrical activity [3,4]. More recently, continuous measurements of atmospheric CO_2_ concentrations have been viewed as being helpful for the evaluation of ventilation conditions in rooms or buildings, and it has been utilized as guidance to avoid the transmission of severe acute respiratory syndrome coronavirus 2 (SARS-CoV-2) [5]. SARS-CoV-2 can cause the coronavirus disease 2019 (COVID-19), which has emerged as a serious problem in respiratory clinical practice [6,7,8]. 

On the other hand, arterial blood gas analysis (ABGA) is very commonly implemented in routine clinical practice of respiratory medicine [9,10,11]. Arterial partial pressure of CO_2_ (PaCO_2_) is commonly evaluated in any type of respiratory disease. PaCO_2_ is useful for the diagnosis of hypo- or hyperventilation and to evaluate potential respiratory drive depression and CO_2_ narcosis in patients with chronic obstructive pulmonary disease (COPD) or other conditions. The evaluation of acid–base imbalance in the context of respiratory acidosis can be performed using pH and PaCO_2_ data. Non-invasive alternative methods such as end-tidal CO_2_ partial pressure of exhaled gas (PetCO_2_) and transcutaneous partial pressure of CO_2_ (PtcCO_2_) have been developed, and their accuracy and usefulness have been evaluated by Bland–Altman analysis [12]. 

Another use of CO_2_ concentration measurements in exhaled air involves assessment of CO_2_ production [9]. The respiratory quotient (RQ) can be calculated using the data of CO_2_ production ($\dot{V}$CO_2_) and oxygen (O_2_) consumption ($\dot{V}$O_2_). Then, the difference of partial pressure of oxygen (PO_2_) between mean alveolar gas and arterial blood can be calculated [10]. This approach has been used for the evaluation of gas exchange impairment in various lung diseases [9,10,13]. Energy expenditure can be also evaluated, and this is particularly of interest in obese patients with obstructive sleep apnea syndrome (OSAS) using CO_2_ production data and oxygen consumption data [14]. 

Thus, depending on the objectives driving the measurement of CO_2_ concentrations, the most suitable method should be adopted. In order to better understand the considerations involved in such choices, we will discuss the principles, sensitivity, and limitations of the various methods available for measuring CO_2_ concentrations.

## 2. Atmospheric Carbon Dioxide Concentration

The World Data Centre for Greenhouse Gases reported that atmospheric CO_2_ concentrations are increasing worldwide, and they are currently around 410 ppm (Figure 1) [2]. The method to measure this concentration is by non-dispersive infrared technology (Figure 2) [15,16,17,18]. This increase in CO_2_ level has been mainly attributed to increasing the consumption of carbon-based energy sources (e.g., coal, oil) with significant concomitant deforestation due to unregulated expansion of industrial agriculture initiatives [1,2].

When atmospheric CO_2_ concentration rises, human PaCO_2_ will rise, but its toxicity has been reported to be little, if any, at 5% (50,000 ppm) or lower [19]. Atmospheric CO_2_ concentrations of more than 50,000 ppm may cause hypercapnia, respiratory acidosis, and increased respiratory rate. Severe acidosis will ultimately result in depression of the respiration and the circulation. Atmospheric CO_2_ concentrations of more than 10% (100,000 ppm) may cause convulsions, coma, and death [19]. 

Duarte et al. showed the standard CO_2_ levels in air in indoors environments (i.e., >15,000 ppm: accident by CO_2_ intoxication; 10,000 ppm: submarines; 5000 ppm: crowded indoors; 600 ppm: well-ventilated indoors) [20].

According to the documents of the World Health Organization, the amplitude (depth) of respiratory movements was reduced by the inhalation of 0.1% (1000 ppm) CO_2_, while peripheral blood flow was increased, and the amplitude of brain electrical waves was increased [3,4]. In these documents from the 1960s, it was suggested that the indoors concentrations of CO_2_ should not exceed 1000 ppm. A man engaged in light work exhales about 22.6 L of CO_2_ per hour [4], and since the recent normal concentration of CO_2_ in the atmosphere is 0.04% (0.4 L/m^3^), the volume of required fresh air per person to ensure CO_2_ concentrations not exceeding 0.1% (1.0 L/m^3^) would be 22.6/(1.0 − 0.4) = 38 m^3^ per hour. Thus, strict monitoring of air circulation and CO_2_ concentrations are essential in indoor locations where the density of humans is high (e.g., cinemas, theaters, office buildings, hospitals, etc.). 

Measuring atmospheric CO_2_ concentrations has been helpful for evaluation of the ventilation conditions in rooms of buildings aiming to decrease the transmission risk of SARS-CoV-2, which can cause COVID-19 (Figure 3) [5,21,22]. Smaller droplets (<10 μm) with SARS-CoV-2 content expired from COVID-19 patients can travel tens of meters in the air while indoors and cause airborne transmission [23,24]. The Japanese government recommended the use of atmospheric CO_2_ sensors in rooms such as restaurants in order to prevent COVID-19 especially in cold weather [25]. Guidelines for indoor CO_2_ concentrations to reduce indoors COVID-19 infection risk should be more helpful if they account for environment and activity types [5]. Marr et al. suggested that indoor CO_2_ concentrations should not exceed 700 ppm in classrooms and 550 ppm in hallways in order to limit the COVID-19 transmission in schools [26]. Teachers in many countries may be required to keep the indoor CO_2_ concentrations low and decrease the students’ risk of inhaling SARS-CoV-2 floating in the air in classrooms. By measuring indoor CO_2_ concentrations, teachers can evaluate how widely the windows should be opened (e.g., fully or partially open) in classrooms considering the meteorological conditions (especially wind) and estimate the overall rate of ventilation in the classroom [26]. 

In addition, there was a fatal accident involving CO_2_ fire extinguishing equipment in Japan in April 2021 [27]. Four people died and two people were injured due to the high concentrations of CO_2_ because the equipment in the basement parking garage was mistakenly activated. The mandate of monitoring atmospheric CO_2_ concentration is increasing and is likely to become mandatory in buildings and similar public structures. Currently, the measurement of CO_2_ concentrations using infrared is the fastest method to obtain data from atmospheric samples at low cost; as such, this method is suitable in most of the situations. 

## 3. Blood Gas Analysis: Principle of PaCO_2_ Electrode

Apart from atmospheric CO_2_ concentration measures, it is frequently necessary to measure the partial pressure of CO_2_ (PCO_2_) in blood in respiratory clinical practice. The analysis of blood gas values has been performed by means of electrochemical devices [28]. The traditionally used electrode for measuring PCO_2_ is termed the Severinghaus PCO_2_ electrode, per the last name of the inventor of this electrode, Dr. John Severinghaus (Figure 4) [28,29]. This PCO_2_ electrode contains the CO_2_-permeable membrane and the principle of pH meter with a pH-sensitive glass membrane. PaCO_2_ is usually measured for the evaluation of any type of lung disease [9,10]. PaCO_2_ is useful for the diagnosis of hyperventilation, hypoventilation, CO_2_ retention, and CO_2_ narcosis in patients with COPD and many other pulmonary conditions [10,30,31].

The evaluation of acid–base imbalance (i.e., respiratory acidosis, respiratory alkalosis, metabolic acidosis, and metabolic alkalosis), with the consideration of compensation, can be performed using simultaneous arterial pH and PaCO_2_ measurements [32,33]. The majority of CO_2_ is transported in the body as bicarbonate ion (HCO_3_^−^) [34]. HCO_3_^−^ plays a central role in maintaining the pH level in blood [32,33,34]. Therefore, it is important to calculate its concentration ([HCO_3_^−^]) in blood using the Henderson–Hasselbalch equation. [HCO_3_^−^] is calculated using the following equation on devices such as Rapidlab 1265 (Siemens Healthcare Diagnostics, Sudbury, UK).
[HCO_3_^−^] = 0.0307 × PCO_2_ × 10^(pH−6.105)^

The normal ranges for PaCO_2_, arterial pH, and arterial [HCO_3_^−^] are 35–45 mmHg, 7.35–7.45, and 22–26 mEq/L, respectively [35]. These data are useful for the calculation of anion gap (AG) [32,34,36]. Using the plasma sodium concentration ([Na^+^]) and plasma chloride concentration ([Cl^−^]), AG is calculated by the following equation.
AG = [Na^+^] − ([Cl^−^] + [HCO_3_^−^])

The normal range for AG is 6–12 mmol/L [32]. AG is utilized for the differential diagnosis of metabolic acidosis. High-AG metabolic acidosis due to increased fixed acid includes ketoacidosis, lactic acidosis, renal failure, toxin by salicylates, etc. [32,34,36]. Normal-AG metabolic acidosis includes renal tubular acidosis, HCO_3_^−^ loss from the gastrointestinal tract, etc. [32,34,36].

The usual clinical practice for ABGA in conscious patients involves a single arterial puncture; however, the procedure may cause pain and cause hyperventilation [11]. PaCO_2_ via the arterial puncture performed after a resting period of 20–30 min has been understood as the gold standard, because arterial blood samples must be drawn when the patient is in a steady state [11,37]. Therefore, newly developed surrogates should be compared with this gold standard PaCO_2_ data.

PaCO_2_ is also useful for the evaluation of the ventilatory support being provided to patients with respiratory insufficiency [38]. However, an arterial puncture is necessary for measuring PaCO_2_, and it is sometimes difficult and painful, e.g., for pediatric patients. Therefore, less invasive or non-invasive surrogate measurements have been sought, and they include venous or capillary partial pressure of CO_2_, PetCO_2_, and PtcCO_2_.

## 4. Non-Invasive Alternative Methods to Estimate PaCO_2_

### 4.1. Venous Blood Gas Analysis (VBGA)

The pulse oximeter allows the measurement of the levels of systemic O_2_ by determining the degree of percutaneous O_2_ saturation (SpO_2_) [39,40]. Therefore, peripheral VBGA with simultaneous evaluation of SpO_2_ offers an alternative to arterial blood gas analysis [41,42,43]. This approach has become standard practice, particularly among pediatric patients and in the emergency department, owing to its advantages (i.e., easiness and less invasive nature) over arterial blood gas analysis [44,45,46]. Capillary blood gas analysis can also be performed. This is particularly useful in children and involves warming the extremity to arterialize the subcutaneous vascular bed and extracting a minute amount of blood using a lancet. The gas content of this sample should be similar to the values obtained for actual arterial blood samples [47,48,49]. It has been demonstrated that intentional hyperventilation increases venous–arterial PCO_2_ differences and pH differences [50]. Moreover, in patients with respiratory alkalosis who did not receive treatment, the condition may be underestimated by the “SpO_2_ plus VBGA” method [50]. Furthermore, hyperventilation increases differences in the concentration of venous–arterial bicarbonate [51]. Therefore, these changes may be attributed to a reduction in peripheral blood perfusion induced by hyperventilation-associated systemic vasoconstriction [50,51]. 

### 4.2. End-Tidal PCO_2_

Traditionally, the concentration of CO_2_ in an exhaled gas is calculated by determining the levels of chemically absorbed CO_2_ and other gases [52,53,54]. The absorbed CO_2_ is subsequently compared with the total volume of the gas, thereby revealing the levels of CO_2_ present. The concentration of CO_2_ in an exhaled gas can also be measured by gas chromatography and/or mass spectrometry, but these systems are voluminous, sturdy, and expensive [55,56,57]. The technological advancement of exhaled CO_2_ monitoring has enabled the reduction of system size and the adequate monitoring of ventilation using the infrared analyzer. PetCO_2_ is the highest and closest estimate of PaCO_2_ in the time course of continuous sampling of expiratory PCO_2_ data [54,58]. Typically, PaCO_2_ and PetCO_2_ differ by 2–5 mmHg. However, the presence of lung disease, such as acute respiratory distress syndrome, COPD, and asthma, ventilation/perfusion ($\dot{V}$/$\dot{Q}$) mismatch (especially relative increase in high $\dot{V}$/$\dot{Q}$ regions) in the lungs can cause the PaCO_2_–PetCO_2_ difference to increase, in which case the non-invasive measurements may be potentially misleading. Patients with gas exchange impairments may be unable to efficiently exhale CO_2_. Therefore, PetCO_2_ is not a good surrogate of PaCO_2_ for patients with pulmonary diseases. Furthermore, PetCO_2_ cannot replace PaCO_2_ [58,59]. Nevertheless, PetCO_2_ has been reported to be a useful indicator of pulmonary perfusion and cardiac output during cardiopulmonary resuscitation [54,58,59,60], and its use was recommended by numerous guidelines (American Heart association [61], European Resuscitation Council [62], and American College of Emergency Physicians [63]). Particularly, the use of waveform capnography was recommended during cardiopulmonary resuscitation [59,61,62]. The return of spontaneous circulation is indicated by a sudden continuous rise in PetCO_2_ (≥40 mmHg) [61]. Patients with an average PetCO_2_ of 15 mmHg are more likely to be successfully resuscitated than those with a value of 7 mmHg [64]. In patients with a low or decreasing PetCO_2_, reassessment of cardiopulmonary resuscitation is recommended [61]. In adults and children, capnometry or capnography can be utilized to continuously monitor alterations in exhaled CO_2_ from the onset of intubation to extubation [54,58,65,66]. Both PetCO_2_ and (PaCO_2_–PetCO_2_) are useful for monitoring $\dot{V}$/$\dot{Q}$ mismatch especially (physiologic deadspace)/(tidal volume) evaluation, and useful to assess pulmonary embolism [58,59]. PetCO_2_ monitoring is a faster indicator than pulse oximetry or ECG tracing in order to find patient mishaps such as a ventilator becoming disconnected or other catastrophic events [58]. 

Monitoring with capnography is recommended not only in intubated patients but also in non-intubated patients undergoing non-invasive positive pressure ventilation (NPPV) [67]. Figure 5 shows the new CO_2_ sensor, TG-980P (Nihon Kohden, Tokyo, Japan) and a mask, cap-ONE (Nihon Kohden, Tokyo, Japan) in the NPPV system with the recently rolled out ventilator, NKV-330 (Nihon Kohden, Tokyo, Japan). In cap-ONE, the inner cup is included, and exhaled air will efficiently reach TG-980P. Monitoring with capnography is possible at a remote place. The electromechanical response of the new devices for NPPV (NKV-330 with cap-ONE and TG-980P), as shown by breathing on the sensor measuring atmospheric PCO_2_, elicited an increase in PCO_2_ within 3 s even at remote places such as a nurse station in a hospital ward. 

There are two methods to sample and detect CO_2_ in clinical situations: mainstream and sidestream [57,68]. Mainstream CO_2_ is measured using a sensor inserted in an airway adapter, and the sample is directly taken from the airway, providing accurate data. Sidestream CO_2_ is measured by pulling the patient’s exhalation air through a small tube into a CO_2_ detector that is placed at the end of the small tube. Although mainstream CO_2_ measurement requires a relatively large amount (150 mL/min) of sample gas, only a small amount (50 mL/min) of gas is sufficient for sidestream [68]. Currently, TG-980P is the smallest and the lightest mainstream PetCO_2_ sensor, where special anti-fog film is used on the window of specimens, and therefore, the heater to avoid fog is unnecessary (Figure 6).

### 4.3. Transcutaneous Blood Gas Analysis

Evaluation of dissolved gases diffusing into the surface of the skin can be used to determine the partial pressure of gases in blood [69,70,71,72,73]. Heating of the skin locally, occasionally accompanied by measurement of transcutaneous PO_2_, is necessary for determining the PtcCO_2_. This dilation of vessels increases the flow of arterial blood to the skin capillary bed below the detector, thereby accelerating the diffusion of gas [69,70,74,75] (Figure 7). According to Severinghaus et al., the PtcCO_2_ electrode contains a relatively large solid silver reference electrode inside the glass pH sensor, which enhances the transfer of heat from the heater to the skin via the glass pH electrode [69,70]. The presence of an ultra-thin film of buffer electrolyte between the silver and glass appeared to be important. This internal electrolyte contains reference solution (e.g., phosphate buffer) (light green, Figure 4 and Figure 7). The external electrolyte contains bicarbonate solution (light blue, Figure 4 and Figure 7). The precise blueprints of recent PtcCO_2_ sensors are different according to manufacturing companies. This approach is commonly used to evaluate the pulmonary gas exchange function in pediatric patients as well as in adults with acute/chronic respiratory failure [76,77,78]. Moreover, this methodology can be employed to monitor patients receiving mechanical ventilation and managing limb ischemia [79,80,81]. 

### 4.4. Comparison of Accuracy

The accuracy of an alternative new method has been evaluated by Bland–Altman analysis for use in respiratory clinical practice (Table 1) [12,45,50,82,83,84,85,86,87,88,89].

PaCO_2_, arterial partial pressure of CO_2_; PetCO_2_, end-tidal CO_2_ partial pressure of exhaled gas; PtcCO_2_, transcutaneous partial pressure of CO_2_; PvCO_2_, venous partial pressure of CO_2_; SD, standard deviation. The width of ± 1.96 SD means the 95% limits of agreement.

## 5. Usefulness and limitation of Transcutaneous Blood Gas Analysis

Currently, the most accurate non-invasive alternative surrogate of PaCO_2_ is PtcCO_2_ (Table 1). We performed various subgroup analyses on the PtcCO_2_ bias (PtcCO_2_—PaCO_2_) in order to use PtcCO_2_ efficiently in the future [89].

### 5.1. Various Subgroup Analyses on the PtcCO_2_ Bias

Subgroup analyses (sex, age, PaCO_2_ level, and PaO_2_ level) were performed using the data at 30 min after the placement of detectors (n = 272).

#### 5.1.1. Sex

The results of the analysis did not show significant differences in the PtcCO_2_ bias (males/females: 168/104 [89]).

#### 5.1.2. Age

Comparison of the PtcCO_2_ bias between four age groups: 20–39 years (n = 11); 40–59 years (n = 12); 60–79 years (n = 138); and ≥80 years (n = 111) (Figure 8a). The PtcCO_2_ bias was significantly lower in young adults (20–39 years) versus those aged 40–59 years and ≥80 years (*p* < 0.05, respectively). PtcCO_2_ and PtcO_2_ are frequently utilized in newborns. The increases in PtcCO_2_ bias induced by aging may be due to the thickness of the skin with increasingly reduced permeability to gas exchange.

#### 5.1.3. PaCO_2_ Level

Comparison of the PtcCO_2_ bias between the severe hypocapnia group (PaCO_2_ < 31 mmHg; n = 7), mild hypocapnia group (31 mmHg ≤ PaCO_2_ < 35 mmHg; n = 24), and normal range group (35 mmHg ≤ PaCO_2_ ≤ 45 mmHg; n = 202) is shown in Figure 8b. The PtcCO_2_ bias was significantly higher in the severe hypocapnia group versus the normal range group (*p* < 0.01), and this was an intensity-dependent effect. Comparison of bias between the normal range group (35 mmHg ≤ PaCO_2_ ≤ 45 mmHg; n = 202), mild hypercapnia group (45 mmHg < PaCO_2_ ≤ 50 mmHg; n = 26), and severe hypercapnia group (50 mmHg < PaCO_2_; n = 13) is shown in Figure 8c. The PtcCO_2_ bias was significantly lower in the mild hypercapnia group versus the normal PaCO_2_ group (*p* < 0.01). The hypocapnic systemic vasoconstriction is thought to be the mechanism of increases in the PtcCO_2_ bias [50]. CO_2_ concentration in blood is very important for peripheral blood perfusion. On the other hand, severe hypercapnic subjects (>50 mmHg) frequently have comorbid conditions such as circulatory failure, heart failure, edema, infection, etc.

#### 5.1.4. PaO_2_ Level

Comparison of the PtcCO_2_ bias between the hypoxemia group (PaO_2_ < 80 mmHg; n = 158), normal range group (80 mmHg ≤ PaO_2_ ≤ 100 mmHg; n = 102), and hyperoxemia group (100 mmHg < PaO_2_, n = 12) is shown in Figure 8d. The PtcCO_2_ bias was significantly lower in the hypoxemia group versus the normal PaO_2_ group (*p* < 0.05), and this was thought to be a PaO_2_ level-dependent effect. Previous studies have investigated hypoxemic systemic vasodilation [90]. The concentration of O_2_ in blood appears to be associated with peripheral perfusion and PtcCO_2_ bias. 

#### 5.1.5. Among Various Respiratory Diseases 

There were not significant differences in the PtcCO_2_ bias among various respiratory diseases in the data of [89] (Figure 9). The breakdown of respiratory diseases was as follows: asthma–COPD overlap (n = 39), COPD due to emphysema (n = 25), interstitial lung disease (n = 41), pneumonia (n = 74), asthma (n = 27), lung cancer (n = 10), acute bronchitis (n = 15), bronchiectasis (n = 7), sleep apnea syndrome (n = 6), pleural diseases (n = 5), and others (n = 15).

### 5.2. Usefulness

The use of this non-invasive PtcCO_2_ monitor leads to an accurate assessment of CO_2_ retention. All hypercapnia patients with PaCO_2_ > 50 mmHg (n = 13→20) showed PtcCO_2_ ≥ 50 mmHg until 12 min [89] (additional data). Utilization of thinner films for CO_2_-permeable and/or pH-sensitive membranes (Figure 7) may accelerate the speed to equilibration in order to reach the accurate data. The American Association for Respiratory Care has recommended an acceptable clinical range of agreement between PtcCO_2_ and PaCO_2_ (±1.96 standard deviation: ±7.5 mmHg or narrower) [80]. This range of agreement, determined through TCM4 with a tcSensor 84 (Radiometer Medical AsP, Copenhagen, Denmark), was reduced over time: ±13.6 mmHg at 4 min, ±7.5mmHg at 12–13 min, and ±6.3 mmHg at 30 min [89].

### 5.3. Limitations

Although PtcCO_2_ is currently the best non-invasive surrogate of PaCO_2_, there were still some cases with large bias over 10 mmHg. PaCO_2_ cannot be replaced with PtcCO_2_ completely even after considering the average bias of 4–5 mmHg (Table 1) [89]. Other limitations include the occurrence of technical drift; therefore, the baseline calibration is necessary [91,92]. In addition, rapid results are not available, and the results are not independent of dermal perfusion, edema, or increased skin thickness [91,93]. 

### 5.4. Future Use

PtcCO_2_ monitoring during sleep study has been reported to be useful for evaluating the necessity of ventilatory support especially in patients with neuromuscular disorders [94,95]. PtcCO_2_ monitoring with polysomnography may become the standard method of sleep study in the future [94]. PtcCO_2_ monitoring during rehabilitation may be the promising method, too [96,97,98]. However, the actual PaCO_2_ will not be disregarded, because the PCO_2_ bias is sometimes large, and PtcCO_2_ cannot replace PaCO_2_ completely [89]. Therefore, future use of PtcCO_2_ monitoring will be limited and may be just focusing on relative evolution.

## 6. Other Applications of Measuring CO_2_ Mainly for Research Use

Measuring CO_2_ in exhaled gas is also used for assessment of the metabolic condition of subjects. Energy expenditure (EE) is determined using the Weir equation (e.g., MK-5000, Muromachi Kikai, Tokyo, Japan) (Figure 10) [99,100,101]. RQ is calculated using the pulmonary exchange ratio ($\dot{V}$CO_2_/$\dot{V}$O_2_)
$$\mathrm{EE}\;(\mathrm{kcal}/\mathrm{kg}/h)\;=\;(3.815\;\times \;\dot{V}O_{2})\;+\;(1.232\;\times \;\dot{V}CO_{2})\;=\;[3.815\;+\;(3.815\;\times \;\mathrm{RQ})]\;\times \;\dot{V}O_{2}$$

The measurement of $\dot{V}$CO_2_ and $\dot{V}$O_2_ is based on the principles of infrared analysis [15,16] and magneto-electrical analysis [102], respectively. The administration of nasal continuous positive airway pressure (CPAP) in patients with sleep apnea has been linked to body weight gain [103,104]. Therefore, long-term exposure to intermittent hypoxia may result in greater reductions in O_2_ consumption and EE. Human and animal studies have examined the metabolic rates. However, the EE or metabolic rates were not found to be decreased in animal models of intermittent hypoxia or in OSAS patients compared to after the treatment with nasal CPAP [14,100]. Conversely, Tachikawa et al. reported significant decreases in basal metabolic rate in OSAS patients by nasal CPAP [14]. Non-agitated sleep without airway obstruction enabled by treatment with CPAP may contribute to this phenomenon. Measure of EE and the calculation of RQ by the pulmonary exchange ratio will undoubtedly contribute to obesity research and other research focused on lifestyle-related diseases in the future [100,105,106,107]. 

When carbohydrates, fat, and protein are oxydized, RQ are calculated to 1.0, 0.7, and 0.8, respectively [108]. Recently, Lin et al. monitored both CO_2_ and O_2_ concentrations in human breath samples using a home-made gas chromatography/milli-whistle analyzer and reported that the changes in CO_2_ concentrations (and the index of CO_2_/O_2_ ratio) were related to the changes in blood sugar concentrations [109]. They sugested that their compact gas chromatography system may be used for a non-invasive and time-dependent (continuous and rapid) blood sugar monitoring in the future. 

In addition, historically, gas chromatography and mass spectrometry had been often used as the gas analyzer in respiratory research, and the peak expired PCO_2_ had been measured by this technology [55,56,57]. The advantage of these methods over the infrared CO_2_ analysis is that concentrations of multiple gases can be simultaneously measured. Nevertheless, the use of mass spectrometry for respiratory research has decreased since 2000, which is likely because of cost and tehcnical fragility of the mass spectrometers, which require more extensive technical support [57]. Furthermore, the method of photoinduced electron transfer is rapidly developping in various research fields, and CO_2_ has been reported to be detected using amine-containing fluorophores [110,111]. The evaluation of local CO_2_ concentrations in various small organs of animals might be possible by this technology. 

## 7. Conclusions

In summary, measures of CO_2_ concentrations in the air are done using the infrared analyzer. Data are important for both the climate problem and the regulatory monitoring of buildings to avoid poor aeration and more recently COVID-19 transmission. Measure of arterial CO_2_ concentration is performed by measuring PaCO_2_ using the Severinghaus electrode. The most accurate non-invasive alternative method of PaCO_2_ is PtcCO_2_. Measure of CO_2_ production with O_2_ consumption may be used for further investigation in the various fields of metabolism, obesity with obstructive sleep apnea syndrome, and lifestyle-related diseases.

## Figures and Tables

**Figure 1 sensors-21-05636-f001:**
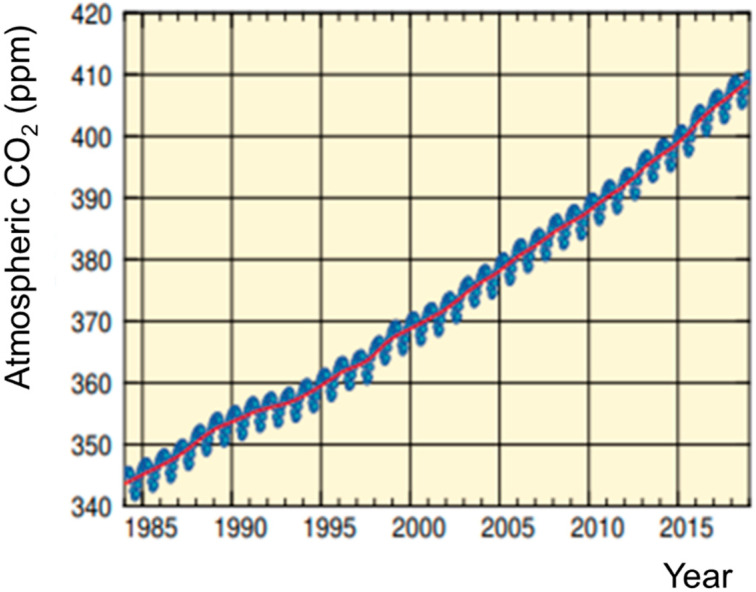
Globally averaged monthly mean mole fraction of CO_2_ from 1984 to 2018 and the deseasonalized long-term trend shown as a red line (Adapted with permission from Ref. [2]. Copyright 2020 WMO WDCGG).

**Figure 2 sensors-21-05636-f002:**
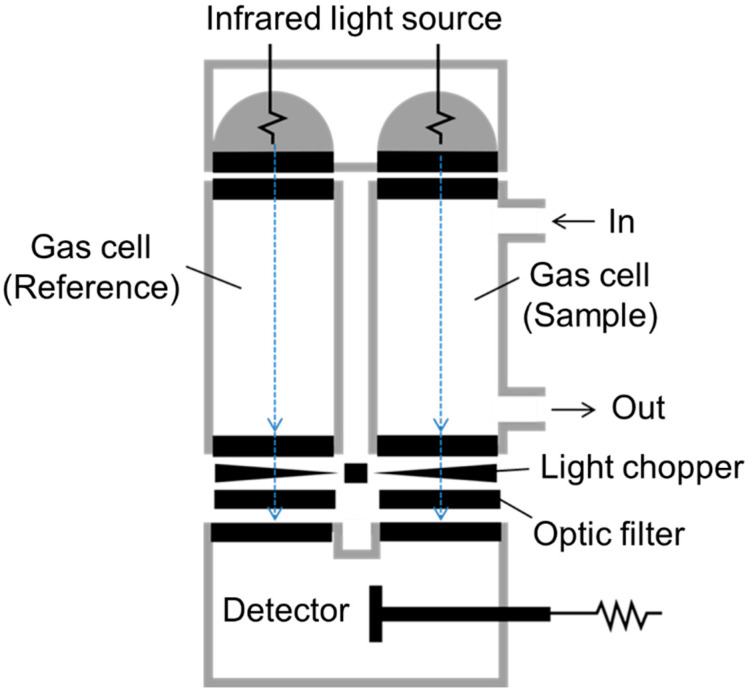
Measuring system of CO_2_ by using the non-dispersive infrared analyzer. The light chopper delivers the data of infrared intensity as a continuous alternating current signal to the detector through the optic filter (Adapted with permission from Ref. [18]. Copyright 2021 HORIBA).

**Figure 3 sensors-21-05636-f003:**
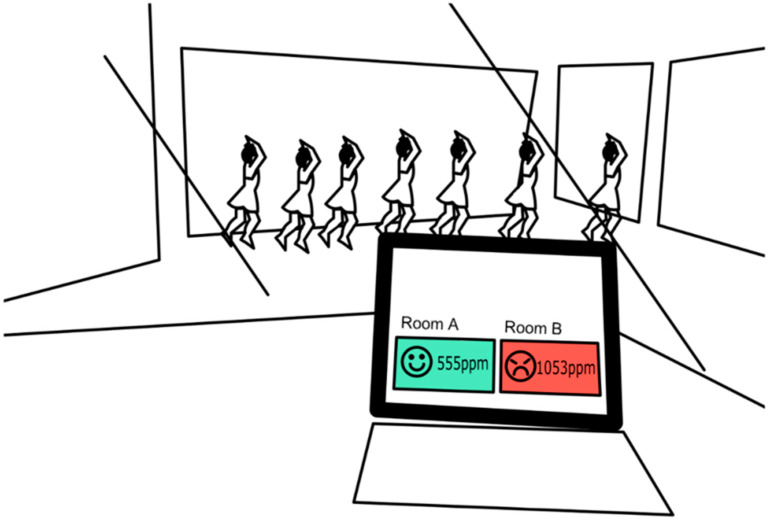
Monitoring of CO_2_ levels in rooms. Higher levels of CO_2_ in a room can mean there is a greater risk of viral transmission (Adapted with permission from Ref. [21]. Copyright 2020 Kyodo).

**Figure 4 sensors-21-05636-f004:**
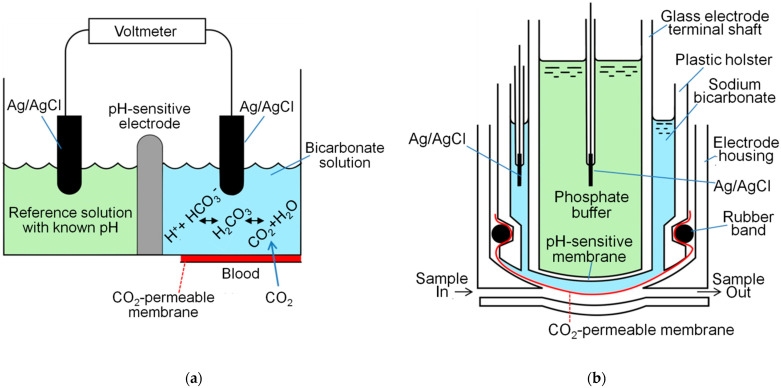
PCO_2_ electrode. (**a**) The CO_2_ from the blood diffuses through the membrane (red) into the bicarbonate solution (light blue). The hydrolysis reaction occurs in the bicarbonate solution and results in the production of hydrogen ions (H^+^) in proportion to the amount of dissolved CO_2_ present. The difference in voltage between the reference solution (light green) and the bicarbonate solution (light blue) is measured. Ag/AgCl, Silver electrode plated with silver chloride; HCO_3_^−^, Bicarbonate ion; H_2_CO_3_, Carbonic acid (Adapted with permission from Ref. [28]. Copyright 2005 Elsevier). (**b**) Severinghaus PCO_2_ electrode. The principle of pH meter with pH-sensitive glass membrane is used (Adapted with permission from Ref. [28]. Copyright 2005 Elsevier).

**Figure 5 sensors-21-05636-f005:**
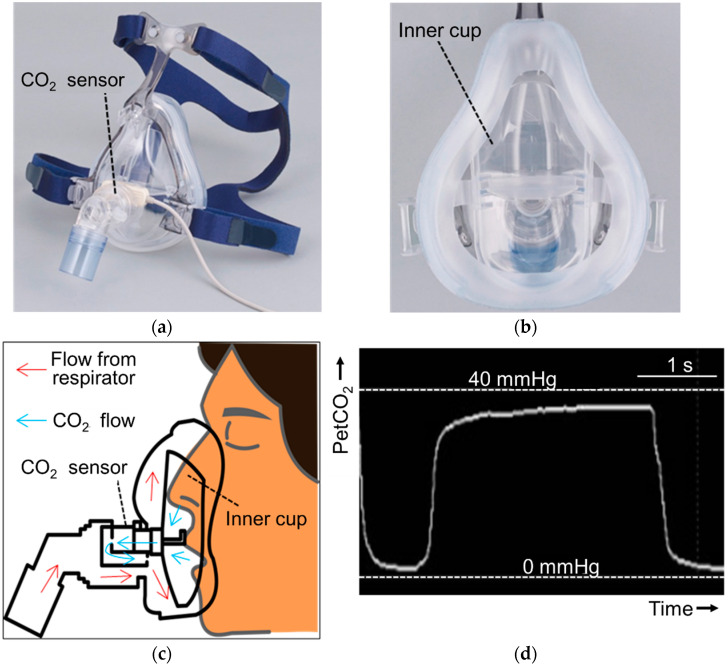
A system of measuring end-tidal PCO_2_ during non-invasive positive pressure ventilation. (**a**) Mainstream CO_2_ sensor (TG-980P, Nihon Kohden, Tokyo, Japan) is used in a mask (cap-ONE, Nihon Kohden, Tokyo, Japan). (**b**) The inner cup is attached inside the mask. (**c**) Air flow from respirator and exhaled flow from mouth or nose are shown. (**d**) Capnographic waveform on the monitor of non-invasive positive pressure ventilator (NKV-330, Nihon Kohden, Tokyo, Japan) is shown.

**Figure 6 sensors-21-05636-f006:**
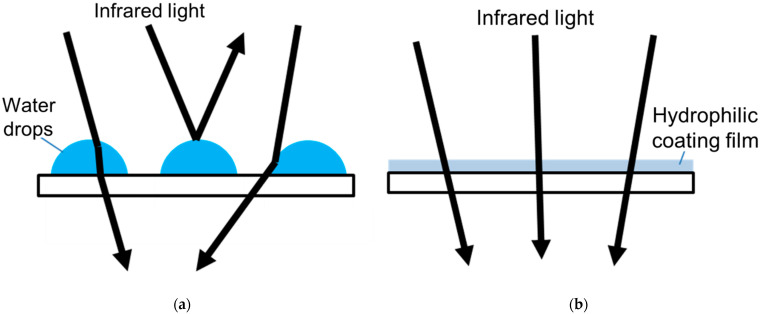
Technology of size reduction of end-tidal PCO_2_ sensors. In the new CO_2_ sensor, the use of heaters to avoid water drops is unnecessary. (**a**) In ordinary CO_2_ sensors, heaters are necessary to prevent windows from being clouded by water vapor in expired air. Water drops on windows cause refraction and reflection of infrared lights. (**b**) The hydrophilic coating film used in the new CO_2_ sensor (TG-980P, Nihon Kohden, Tokyo, Japan) disabled the surface tension of water drops. Thanks to this anti-fog film, the new sensor does not require the use of heaters.

**Figure 7 sensors-21-05636-f007:**
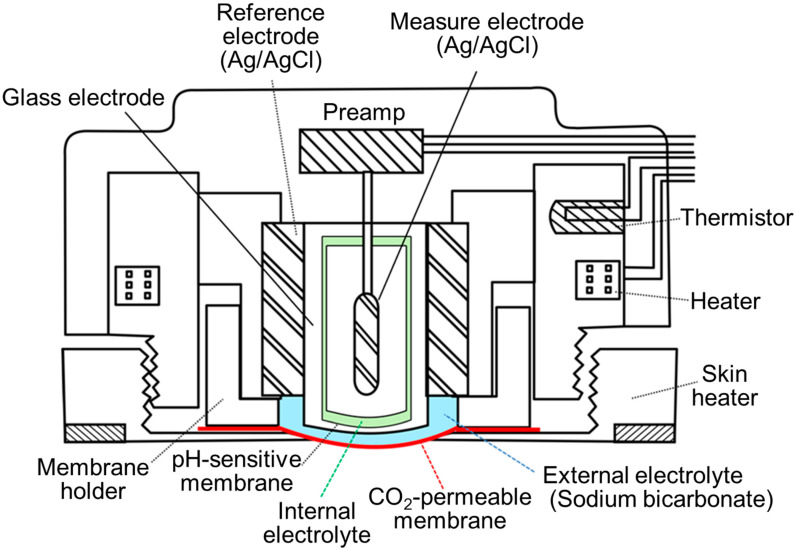
Transcutaneous PCO_2_ sensor. The skin heater is necessary in addition to the Severinghaus electrode (Adapted with permission from Ref. [73]. Copyright 1983 Japanese Society for Medical and Biological Engineering). An ultra-thin film of buffer electrolyte (light green) is placed between the silver and glass. This internal electrolyte stabilizes the pH inside the glass electrode. The external electrolyte (light blue) contains sodium bicarbonate. According to the manufacturing companies, the precise blueprints of recent products differ.

**Figure 8 sensors-21-05636-f008:**
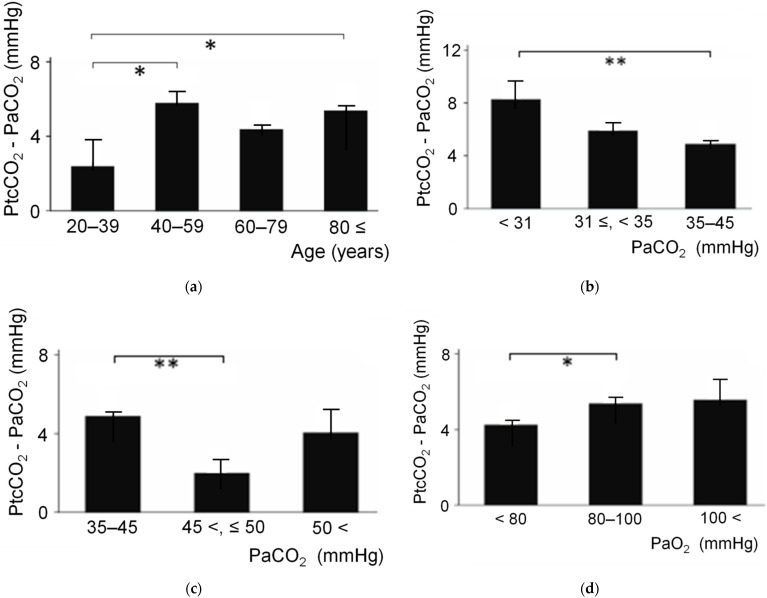
Comparisons of PtcCO_2_ and PaCO_2_ bias (n = 272). (**a**) Comparison of bias between four age groups. The bias was significantly lower in young adults (20–39 years) versus those aged 40–59 years and ≥80 years. (**b**) Comparison of bias between the severe, mild hypocapnia group, and normal range group. The bias was significantly higher in the severe hypocapnia group than the normal range group, and this was an intensity-dependent effect. (**c**) Comparison of bias between the normal range group and mild, severe hypercapnia group. The bias was significantly lower in the mild hypercapnia group versus the normal range group. (**d**) Comparison of bias between the hypoxemia group, normal range group, and hyperoxemia group. The bias was significantly lower in the hypoxemia group versus the normal range group, and this was a PaO_2_ level-dependent effect. Bars: SEM, *: *p* < 0.05, **: *p* < 0.01 [89]. PaCO_2_, arterial partial pressure of CO_2_; PaO_2_, arterial partial pressure of O_2_; PCO_2_, partial pressure of CO_2_; PtcCO_2_, transcutaneous partial pressure of CO_2_; SEM, standard error of the mean.

**Figure 9 sensors-21-05636-f009:**
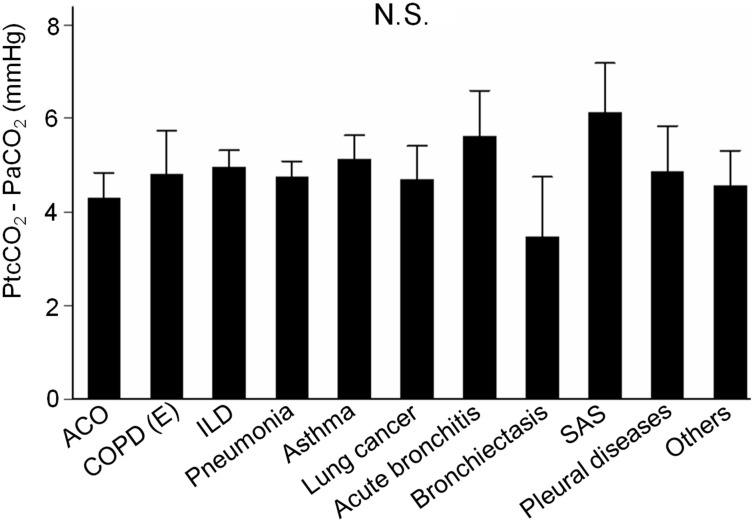
Subgroup analyses on PCO_2_ bias (PtcCO_2_—PaCO_2_) of patients with various respirtory diseases (n = 272). Bars: SEM. There were no significant differences in PCO_2_ bias (ANOVA with Tukey’s post hoc test). ACO, asthma-chronic obstructive pulmonary disease overlap; ANOVA, analysis of variance; COPD, chronic obstructive pulmonary disease; E, emphysema; ILD, interstitial lung disease; N.S., not significant; PaCO_2_, arterial partial pressure of CO_2_; PCO_2_, partial pressure of CO_2_; PtcCO_2_, transcutaneous partial pressure of CO_2_; SAS, sleep apnea syndrome ([89], additional data).

**Figure 10 sensors-21-05636-f010:**
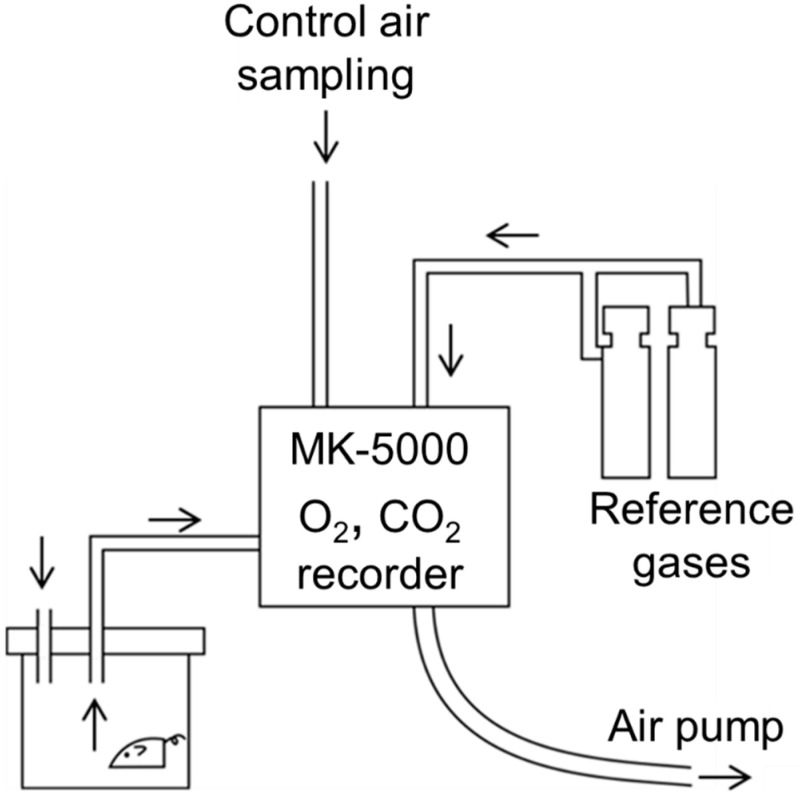
The system of measuring CO_2_ production and O_2_ consumption in mice. Inlet air routes from animal chamber, control air, and reference gases are periodically changed. CO_2_ concentration is measured by infrared absorption analysis. O_2_ concentration is measured by magneto-electrical analysis (MK-5000) [100].

**Table 1 sensors-21-05636-t001:** Alternative non-invasive methods for measuring PaCO_2_.

Surrogate	Average Bias	1.96 SD	Accuracy	Usefulness for Patients with Pulmonary Diseases	References
PvCO_2_	Approximately 5 mmHg higher than PaCO_2_	14.7–15.0 mmHg	Worst	Limited	[45,50]
PetCO_2_	2–5 mmHg lower than PaCO_2_	6.9–14.4 mmHg	Second best	Limited	[83,84,85,86,87,88]
PtcCO_2_	4–5 mmHg higher than PaCO_2_	4.6–10.4 mmHg	Best	Good (still not replaceable)	[83,84,85,86,87,88,89]

## Data Availability

Not applicable.

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
