# Peer review of "Recent Insights into the Measurement of Carbon Dioxide Concentrations for Clinical Practice in Respiratory Medicine"

_sensors, 2021, doi:10.3390/s21165636_

Round 1

Reviewer 1 Report

Congratulations on the review, I indicated some suggestions in the text. Look carefully to make the article more fluid.

Reviewer 2 Report

the review is poorly organized, only section 4 descibed the measurement of CO2 in  respiratory, and the methods are not new and state of art, other parts of paper are not related to the title.

Many methods have been applied to measure CO2 in respiratory, the authors don't even try to review them.

Author Response

Dear Reviewer 2,

Thank you very much for reviewing our article entitled, “Recent insights into the measurement of carbon dioxide concentrations in the field of respiratory medicine”.

Comments: the review is poorly organized, only section 4 descibed the measurement of CO2 in respiratory, and the methods are not new and state of art, other parts of paper are not related to the title. Many methods have been applied to measure CO2 in respiratory, the authors don't even try to review them.

Answer: There seemed to be a great misunderstanding. This title reminded you that CO2 measurement for all fields of respiratory medicine, I think. What we want to write most is the topic on the “clinical use of CO2 measurement” for usual respiratory clinical practice. Methods for research use in respiratory medicine are additional and we wrote these in Section 6 with the title of “… mainly for research use”. Therefore, we would like to change the title to “Recent insights into the measurement of carbon dioxide concentrations for clinical practice in respiratory medicine”. We think this title is much better. Related portions were continuously changed. New information was added to each section. Especially, CO2 detection by “photoinduced electron transfer” in Section 6 was surprising for me (Revised manuscript Line 531-535). Thank you very much for giving us the opportunity to review literature again.

Thanks to your comments, we could change the title to a much better one. Thank you very much indeed.

Sincerely,

6 Aug 2021

Akira Umeda, M.D., Ph.D.  

Reviewer 3 Report

The detection of carbon dioxide is very interesting in the field of respiratory medicine, although the importance of measuring carbon dioxide (CO2) concentrations cannot be overemphasized. The paper reviewed many applications of measuring CO2, including invasive methods and non-invasive methods. It also introduced the principle, merits and flaws of the CO2 measurement methods. The paper is suitable to published in the journal.

Author Response

Dear Reviewer 3,

Thank you very much for reviewing our article entitled, “Recent insights into the measurement of carbon dioxide concentrations in the field of respiratory medicine”.

The title was changed to “Recent insights into the measurement of carbon dioxide concentrations for clinical practice in respiratory medicine” and as such also address another reviewer’s comments.

Sincerely,

6 Aug 2021

Akira Umeda, M.D., Ph.D.  

Reviewer 4 Report

This review paper presents an interesting overview of medical and health related applications for CO2 monitoring. The technologies for measuring CO2 are discussed in sufficient detail for the scope of this review. Transcutaneous versus arterial CO2 measurement is discussed and a  subgroup analysis is presented to demonstrate the usefullness of PtrCO2 measurement. It is expected that PtrCO2 will be increasingly used in sleep studies. Besides PtrCO2, also exhaled CO2 and O2 measurements are interesting for the assessment of metabolic condition and energy expenditure.

The paper is well presented and includes relevant citations to recent work.

There are minor textual corrections needed:

page 4, line 110: litters -> liters

page 8, line 254: alter ations -> alterations

Author Response

Dear Reviewer 4,

Thank you very much for reviewing our article entitled, “Recent insights into the measurement of carbon dioxide concentrations in the field of respiratory medicine”.

As indicated in our response to other reviewers, the title was changed to “Recent insights into the measurement of carbon dioxide concentrations for clinical practice in respiratory medicine”.

Indicated misspellings were corrected.

Sincerely,

6 Aug 2021

Akira Umeda, M.D., Ph.D.   

Reviewer 5 Report

Dear authors

The paper presents a review on the measurements of CO2 in atmospheric and in health science. The review makes a good approximation of the different methods and their applications with a special focus on the health science field. The review is interesting and informative and it merits its publication. But before, I would like to comment on a few small changes or clarifications.

Lines:

57: Maybe you could add a reference to any study of ventilation and CO2 monitoring?

93: Maybe a small explanation of the mechanism like in the other figures would be a good addition.

103: What “accident” means in this context?

472: Could be a future use to use PtcCO2 to monitor evolution disregarding the actual PaCO2, just focusing on relative evolution?

521: You make one comment on coronavirus that is not justified. A mention of the PaCO2 contents related to the SARS-CoV-2 could be more justified.

Best regards.

Round 2

Reviewer 2 Report

In my opinion, review should gives readers some new insigh perspectives. This paper reviewed the measurement of CO2 and its application in  respiratory medicine. Therefore, authors should presents some new and novel methodology on CO2 measurement and outlooks its future direction. Moreover, the challenges on CO2 detection are required. For the application part, what's new results and development on breathing CO2?

The paper should reorganizes according to the parts of CO2 detection and its appication in respiratory medicine.